# Rational Engineering of (*S*)-Norcoclaurine Synthase for Efficient Benzylisoquinoline Alkaloids Biosynthesis

**DOI:** 10.3390/molecules28114265

**Published:** 2023-05-23

**Authors:** João P. M. De Sousa, Nuno C. S. A. Oliveira, Pedro A. Fernandes

**Affiliations:** LAQV/REQUIMTE, Departamento de Química e Bioquímica, Faculdade de Ciências Universidade do Porto, Rua do Campo Alegre, s/n, 4169-007 Porto, Portugal

**Keywords:** enzyme catalysis, protein engineering, alkaloid biosynthesis, benzylisoquinoline alkaloids, (*S*)-norcoclaurine

## Abstract

(*S*)-Norcoclaurine is synthesized in vivo through a metabolic pathway that ends with (*S*)-norcoclaurine synthase (NCS). The former constitutes the scaffold for the biosynthesis of all benzylisoquinoline alkaloids (BIAs), including many drugs such as the opiates morphine and codeine and the semi-synthetic opioids oxycodone, hydrocodone, and hydromorphone. Unfortunately, the only source of complex BIAs is the opium poppy, leaving the drug supply dependent on poppy crops. Therefore, the bioproduction of (*S*)-norcoclaurine in heterologous hosts, such as bacteria or yeast, is an intense area of research nowadays. The efficiency of (*S*)-norcoclaurine biosynthesis is strongly dependent on the catalytic efficiency of NCS. Therefore, we identified vital NCS rate-enhancing mutations through the rational transition-state macrodipole stabilization method at the Quantum Mechanics/Molecular Mechanics (QM/MM) level. The results are a step forward for obtaining NCS variants able to biosynthesize (*S*)-norcoclaurine on a large scale.

## 1. Introduction

The BIAs are one of the most relevant drug groups for chronic pain management, whose most well-known members include the opiates morphine, codeine, and thebaine and the semi-synthetic opioids oxycodone, hydrocodone, and hydromorphone [1]. These drugs are listed as essential medicines by the World Health Organization for their utility in managing chronic pain and palliative care [2]. Despite this, more than one-half of the world’s population has low to nonexistent access to treatment for moderate or severe pain [3]. The only source of these drugs or their synthetic precursors is the opium poppy (*Papaver somniferum*) harvest, which depends on the highly-susceptible opium poppy crops, introducing a risk factor in their worldwide supply [4,5]. Biotechnology-based forms of BIAs production arising from the poppy’s enzymes usage in solution or heterologous hosts can be one solution for diversifying BIAs production methods [6,7,8,9,10].

Currently, the enzyme-based alternatives for diversifying production are mostly focused on using either *Escherichia coli* or *Saccharomyces cerevisiae* as microbial hosts for the synthesis of BIAs [11]. The most prominent development of BIAs production was achieved by Pyne and coworkers in *S. cerevisiae* using a strain capable of synthesizing (*S*)-reticuline to titers up to the 4.6 g L^−1^ [8], an improvement of 57.000-fold compared to previous efforts [10]. The pathway engineering was performed by manipulating both endogenous and exogenous genes to maximize carbon flux into BIAs biosynthesis and minimize the formation of fusel products.

The early stages of naturally occurring opioids in the opium poppy are shown in Figure 1. BIAs downstream products have a high degree of three-dimensional structure complexity and might have up to five stereocenters, as in the case of morphine or codeine. Nonetheless, these complex molecules arise from the condensation of two equivalents of L-tyrosine derivatives, dopamine, and 4-hydroxyphenylacetaldehyde (4-HPAA). Endogenous l-tyrosine can be first hydroxylated to L-DOPA and then *P. somniferum*’s aromatic amino acid decarboxylase (AADC) decarboxylates L-DOPA to synthesize dopamine. Alternatively, the *P. somniferum* AADC can also decarboxylate L-tyrosine to tyramine, which can be further hydroxylated to dopamine. Another L-tyrosine equivalent is necessary to synthesize 4-HPAA; an L-tyrosine aminotransferase (TyrAT) converts the L-tyrosine residue to 4-hydroxyphenylpyruvate (4-HPP), which is then decarboxylated to yield 4-HPAA. Besides the naturally occurring opioids, (*S*)-norcoclaurine is also the precursor to other bioactive BIAs such as magnoflorine, sanguinarine, and noscapine [1].

Next to the derivatization of the two L-tyrosine equivalents, NCS comes into play, one of the most relevant enzymes in the BIAs biosynthesis pathway. This enzyme is responsible for the enantioselective condensation of dopamine and 4-HPAA (Figure 1). Since NCS has two substrates, the order of substrate binding to the active site pocket influences the outcome of the reaction. X-ray structural evidence deposited under the PDB accession code 5NON led to a general acceptance that dopamine is the first substrate to enter deep within the active site of NCS, followed by 4-HPAA which lodges at the active site entrance. This substrate binding order is corroborated by NCS’s high substrate promiscuity towards the aldehyde substrate and the fact that a catalytic lysine residue establishes an interaction with the catechol moiety of the bound mimic, demonstrating this residue’s crucial activity as a proton acceptor during NCS’s reaction mechanism [12]. Other substrate binding poses might exist within the enzyme active site, at certain time periods, as previous crystallographic structures have demonstrated (PDB accession code: 2VQ5), but these alternative poses might lead to unreactive enzyme:substrate complexes and therefore dissociate [13].

The demonstrated promiscuity of NCS opens the possibility of engineering NCS variants with enhanced activity towards synthesizing (*S*)-norcoclaurine [14,15,16].

Furthermore, NCS is considered as a key bottleneck in BIAs biosynthesis once a drop in titer was reported from dopamine to (*S*)-norcoclaurine synthesis (from mg L^−1^ to μg L^−1^) [10].

The viability of heterologous production of BIAs on an industrial scale is dependent on the improvement of the catalytic efficiency of the pathway enzymes to achieve viable product concentrations, space–time yields, and catalyst productivity, which in turn determine the commercial viability of the method [17]. Being NCS the initial branchpoint and a significant bottleneck in the BIAs biosynthesis pathway, it is one of the most appealing targets for biocatalyst improvement [18,19].

Previous structural and computational studies have dissected the intricacies of *Thalictrum flavum* NCS (*Tf*NCS) substrate binding, reaction mechanism, and enantioselectivity [12,20]. The complete cycle involves five transition states, from which the last and rate limiting is the rearomatization triggered by the deprotonation of the quinonoid intermediate (QIN) at the C5 position, which leads to the protonation of the quinone moiety by the catalytic Lys residue (Figure 2).

In this work, we have used the ‘*own n-layered integrated molecular orbital and molecular mechanics*’ (ONIOM) QM/MM method [21] to calculate the contribution of the residues outside the active site for the activation energy of the rearomatization step, as was previously performed for other enzymes with biotechnological applications [22,23]. Our approach began with simulating the quinonoid intermediate (QIN) rearomatization step for the wild-type *Papaver somniferum* NCS (*Ps*NCS) through a linear transit scan of the QIN-H•••^−^OOC-Glu109 coordinate. We then fully optimized the reactant (R) and transition state (TS) stationary points for this specific step and calculated the energy barrier for the wild-type *Ps*NCS (Δ*E*_wt_). All the details concerning the modeling of *Ps*NCS and building the QM/MM model can be found in the methods section. Next, a point deletion of each residue from the MM layer in both R and TS was performed, and the associated energy barrier (Δ*E*_m_) was calculated. The magnitude of each residue’s influence in the energy barrier (ΔΔ*E*) is given by the difference in energy between the Δ*E*_m_ and Δ*E*_wt_ ($\Delta \Delta E_{m-\mathrm{wt}}=\Delta E_{m}-\Delta E_{\mathrm{wt}}$). We propose that the experimental procedure to design NCS variants with enhanced first-order reaction rates should focus on residues that raise the wild-type barrier, which can be identified by the ones that retrieve a negative ΔΔ*E*_m−wt_ value.

## 2. Results and Discussion

The generated conformations from the MD simulations reveal an average RMSD of 1.70 Å for the protein backbone and 0.69 Å for the active site (see Appendix A). Therefore, we can infer that QIN and the modeled catalytic water molecule do not move away from their positions in the active site and that the overall structure of *Ps*NCS is stable. This is not only evidence of a well-constructed enzyme model but also of a realistic docking pose for reaction to happen.

We have also monitored the variations in the length of the two most important interactions for the reaction: Lys121-NH3^+^•••O=C-QIN and Glu109-COO^−^•••HC-QIN. The collected data indicate that these are well conserved through the course of the MD simulation (Figure 1 and Figure 2). Regarding the interaction with Glu109, it is interesting to note that its carboxylate side chain has no conformational freedom to switch around the oxygen atoms, as depicted by the graphic. We see that one of them is always closer to QIN’s leaving proton than the other, with an average distance of 2.99 Å against one of 4.80 Å (Figure 1). This is probably due to the hydrogen bonding network established in the active site, which may induce it to adopt specific conformations to favor the course of the reaction. As for the interaction with Lys121, the almost superimposable graphics lead us to deduce that there is no specificity in this protonation. The amino group side chain of Lys121 has the torsional freedom to flip, so all the protons are in range for hydrogen bond formation with the carbonyl group of QIN, with average distances between 2.98 and 3.16 Å (Figure 2).

As the MD simulations show that our model maintains the distances involved in the QIN reactivity, we selected the structure obtained after the phased minimization protocol to proceed to the simulation of the rate limiting step through QM/MM.

The overall orientation of the active site was maintained after initial QM/MM geometry optimizations, even though a slight approximation of QIN towards Lys121 was noticed (Figure 3).

As expected, the TS of the rate-limiting rearomatization step is characterized by the concerted and asynchronous proton transfer triggered by QIN deprotonation at the C5 position.

A closer look at the TS geometry confirms the stepwise fashion of the proton transfers occurring in QIN as the C5 proton is halfway through the QIN-H•••^−^OOC-Glu109 coordinate. In contrast, the bond between the Lys121 amino N and its proton is not yet broken (Figure 3).

The single-point energy calculations employing the ONIOM(B3LYP-D3/6-311+G(2d,2p):amber theoretical level retrieved a ∆*E*_wt_ for *Ps*NCS of 10.6 kcal mol^−1^ in very good agreement (~2 kcal mol^−1^ lower) with the *Tf*NCS value predicted by Himo and Sheng [20].

The influence of the deletion of each residue on the barrier is shown in Figure 4. As the transformation of the R into the TS involves a partial neutralization from the Glu109 and the development of a negative charge at the QIN oxygen atom in the *para* position (QIN_OX_; Lys121 is still positive at the TS), all residues that stabilize the negative charge at Glu109 more extensively than at QIN_OX_ will over stabilize the R in relation to the TS, generating an anticatalytic effect. Therefore, these shall be the principal target residues for rate-enhancing mutations. Thus, negative residues closer to QIN_OX_ than to Glu109 or positive residues closer to Glu109 than to QIN_OX_ are the ideal targets for mutation. This principle is confirmed by the obtained results, highlighted in Figure 4.

In summary, the charged residues that raise the barrier for more than 1 kcal mol^−1^ and thus are the ideal candidates to test for experimental mutagenesis are Asp64, Asp71, Arg105, Lys110, Glu117, and Lys163. The large barrier-lowering effect induced by the deletion of Tyr60 is due to the loss of an ionic interaction with the Lys121 amino group and Tyr60 backbone carbonyl, which destabilizes the reactant. However, for mutation purposes, Tyr60 is not an ideal target because it is an important residue for the previous steps of the NCS mechanism.

The results in Figure 4 hold if the enzyme does not undergo significant structural rearrangements after introducing the mutations. However, experience shows that many mutations cause structural rearrangements. Thus, ideal mutations cancel the side chains charge but retain most of the other characteristics of the residues, such as Glu/Gln, Asp/Asn, or Lys/Gln.

In general, mutating a charged residue for a polar residue is not highly disruptive because the environment around the charged residue is highly polar, responding well to the mutation. Conversely, mutation of a charged/polar residue by a hydrophobic residue, and vice versa, is highly disruptive because the surrounding environment is not tuned for the mutation.

However, in NCS, the charged residues identified as favorable for mutation, although not far from the active center, are located on the enzyme’s surface. Moreover, their charged groups are exposed to the solvent and free of hydrogen bridges with other residues. This is an exceptionally favorable situation for mutation because of the low disruptive propensity of the mutations. The function of these residues likely is to confer adequate solubility to the NCS, so their mutation by polar or oppositely charged residues seems amenable from a structural perspective.

In sum, the proposed mutations are not disruptive nor are significant changes in the protein scaffold expected.

## 3. Methods

The computational protocol in this work can be described as follows: (a) construction of the PsNCS:QIN complex through homology modeling and molecular docking; (b) MD simulations to stabilize the modeled structure (50 + 100 ps) and to briefly sample the conformational space of the system (10 ns); (c) QM/MM calculation of the reactant, transition state, and product of the last, rate-limiting, step of the reaction; and (d) calculation and rationalization of the influence of the residues surrounding the active site on the activation energy and prediction of specific mutations that increase the reaction rate.

### 3.1. Construction of the Psncs:Qin Complex

To the date of this work, there were no solved crystallographic structures of PsNCS available at the Protein Data Bank. To overcome this, we built a PsNCS structure through homology modeling, using SWISS-MODEL [24,25,26,27,28]. After choosing a *Ps*NCS target sequence (UniProtKB accession number Q4QTJ1), the SWISS-MODEL template library (SMTL version 05/05/2021, PDB release 30/04/2021) was searched with BLAST [29] and Hhblits [30,31] for evolutionary related structures matching the target sequence. From the results gathered by this sequence alignment-based search, we chose a recent truncated *Tf*NCS crystallographic structure (PDB ID 6RP3) as the template [32]. It was obtained by X-ray crystallography with a resolution of 1.81 Å and displayed a sequence identity of 53.70%. These are already signs of a potentially precise model, confirmed with our subsequent structure validation tests (results shown in Appendix A). For internal structure validation, we created a Ramachandran plot of our model using MolProbity (Appendix A) [33,34,35]. For external structure validation, we evaluated its tertiary structure (folding) by comparing its structural motifs with the ones found in native proteins of identical size, using the estimators QMEAN, from SWISS-MODEL, and ProSaII, from ProSa-Web (Appendix A) [36,37]. We added the hydrogen atoms to our structure and solvated the enzyme with a cubic box of TIP3P water molecules [38], using the xLEaP program from Amber18 [39]. The system was then subjected to a four-step minimization protocol. In the first step, the water molecules were minimized, while harmonic potentials restrained the other atoms of the system. Afterward, we sequentially allowed the hydrogen atoms, the residue’s side chains, and the whole system to move freely. In each minimization, the restraint force constant applied to the fixed atoms was 50.0 kcal mol^−1^ Å^−2^.

The template structure had a reaction intermediate mimic from which we built the QIN ligand with the GaussView software [40]. For the protein–ligand docking, the solvation box had to be removed from the system. The technique was performed with vsLab [41], a VMD [42] plug-in for molecular docking procedures using the AutoDock software [43,44,45]. For the search stage, we set up a box comprising the area of the active site and defined the number of possible results to 15, using the default options for the genetic algorithm. The scoring stage was carried out with the AutoDock scoring function. From the resulting four poses (A to D, see Appendix A), we chose to proceed with complex B because the position and interactions of QIN were the most appropriate for the reaction to happen.

The parameters for QIN were determined from an optimized structure at the HF/6-31G(d) level in vacuum. We used the antechamber program, from AmberTools18 [39], with the general AMBER force field (GAFF) [46] to derive the intramolecular and Lennard–Jones parameters, and the restrained electrostatic surface potential (RESP) method [47] to derive atomic charges at the HF/6-31G(d) level of theory. We manually added a water molecule to the active site and protonated the carboxylate side chain of the residue Asp140 because these characteristics are considered relevant for the reaction’s mechanism [20]. The AMBER *ff*14SB force field [48] was used to parametrize the remaining atoms of the enzyme. To prepare the system for the simulations, a new cubic box of TIP3P water molecules [38] was added to solvate the enzyme within 10 Å of the solute, as well as 2 Na^+^ and 1 Cl^−^ counterions to neutralize the system’s charge. These procedures were carried out with xLEaP, from Amber18. This system was subjected to a five-step minimization protocol, with the same restraint force constant for the fixed atoms. In the first step, only the manually added water molecule was allowed to move. The remaining water molecules, the hydrogen atoms, the residue’s side chains, and the whole system were sequentially minimized afterward. Both minimizations were carried out with the *sander* module of Amber18.

### 3.2. MD Simulations

The simulated system had 36,079 atoms. The SHAKE algorithm [49] was used in all stages of the simulation to constrain all bonds involving hydrogen atoms, allowing the usage of an integration time step of 2 fs. The particle mesh Ewald (PME) method was used to calculate the long-range interactions, whereas short-range interactions are described with Lennard–Jones potentials, using a 10 Å cutoff.

A 20 ps progressive heating was performed linearly from 0 to 310.15 K, followed by 30 ps at 310.15 K in the canonical ensemble (NVT). The system was equilibrated at that temperature for 100 ps in the isothermal-isobaric ensemble (NPT, 310.15 K, 1 bar), using the Langevin thermostat with a collision frequency of 2 ps^−1^ and isotropic position scaling. A production stage was then run in the NPT ensemble for 10 ns to analyze the conformations of the active site and the stability of the protein backbone, in the same regulation conditions. Heating and equilibration stages were run with the sander module of Amber18, and PMEMD.CUDA [50,51,52] was used for the production stage. The presented root-mean-square deviation (RMSd) values were calculated with CPPTRAJ [53] from AmberTools18. Data on the variations in the length of interactions were acquired from VMD.

### 3.3. Oniom Model Details

The QM/MM model was defined from the minimized structure obtained after the docking procedure. The counterions and most of the solvation box were removed, keeping a solvation shell with all water molecules within 7 Å of the enzyme, ensuring that all the outer residue’s side chains were solvated. The final model had a total of 6706 atoms and an overall charge of −1 au. The QM layer, treated at the density functional theory level, comprised 107 atoms, including the QIN substrate, the catalytic water molecule, the side chains of the proposed catalytic residues (Tyr107, Glu109, Lys121, and Asp140), and the side chain of the nearby interacting residue Tyr138, as depicted by Figure 3. The remaining 6599 atoms of the system were described by the AMBER *ff*10 force field [54] and the MM layer waters were kept fixed.

All QM/MM calculations were carried out within the ONIOM methodology [55,56] using the Gaussian 09 program. Preparation of the input files and output files analysis was facilitated using molUP [57] a VMD plug-in providing a graphical interface for preparation and analysis of QM/MM calculations. The truncated bonds in the QM layer’s frontier were treated with the link-atom approach using hydrogen atoms [58]. The system’s geometry was optimized twice: a first optimization with the mechanical embedding scheme was followed by a second one employing the electrostatic embedding scheme.

In order to study the reaction path, we scanned this optimized structure’s reaction coordinate (i.e., the distance between the oxygen of the carboxylate side chain of Glu109 and the hydrogen of QIN). The structure corresponding to the energy maximum in the scan was taken as a guess for a subsequent full transition state (TS) optimization, and its vibrational frequencies were calculated to confirm its TS nature (a single imaginary frequency). We followed the intrinsic reaction coordinate (IRC) [59] of the TS to obtain the reactants and products of the reaction, which were further optimized and also subjected to a frequency calculation to verify the absence of imaginary frequencies.

Zero-point, thermal, and entropic corrections were also calculated, at 298.15 K, to obtain free energies of activation. In order to assess the influence of the MM layer residues on the activation energy, we performed a series of single-point energy calculations removing every protein residue within 10 Å of the active site region, other than Gly or Pro, employing an approach proven successful in previous work carried out within our group [22,60,61]. The difference in activation energy calculated with a structure with a deleted residue and the complete structure gives the QM/MM energy contribution of that residue for the wild-type barrier.

All QM/MM calculations were performed using the B3LYP density functional [62,63]. The 6-31G(d) basis set [64] was used for the scans, optimizations, and frequency calculations, whereas the larger 6-311+G(2d,2p) basis set, along with the (D3) dispersion correction [65], was used for the single-point calculation of the three optimized stationary states (Appendix A).

### 3.4. Transition-State Macrodipole Stabilization

In order to assess the influence of the MM layer residues on the activation energy, we performed a series of single-point energy calculations removing every protein residue within 10 Å of the active site region, in both R and TS stationary structures, other than Gly or Pro, employing an approach proven successful in previous work carried out within our group [22,60,61]. The difference in activation energy calculated with a structure with a deleted residue and the complete structure gives the QM/MM energy contribution of that residue for the wild-type barrier.

A total of 105 individual residue deletions in the MM layer residues of each stationary structure were performed, adding up to 210 single-point calculations for this experiment. It is important to note that, as a conceptual experiment, the values obtained for each deletion are simply a quantitative estimate of the stabilization or destabilization effect that a specific residue might have on the reaction barrier and are not meant to be a theoretical calculation of the activation free energy associated to an NCS mutant.

In the conceptual deletion experiment, the values analyzed for the energetic differences are the ones obtained from the single-point calculations, without zero-point, thermal, or entropic corrections. The interaction between the layers was treated with the electrostatic embedding formalism.

## 4. Conclusions

The essence of catalysis is the stabilization of the transition state with respect to the reactant. It is useless to predict mutations that decrease the activation energy by looking at a crystallographic structure since this only represents one piece of the puzzle, the reactant state. While crystallographic structures co-crystallized with transition state analogs can be helpful, the fine detail they provide is not sufficient. For example, the protonation of Glu109 might antecede, precede, or be synchronous with the deprotonation of Lys121. Such fine details change the associated transition state’s nature and charge distribution and are intangible by looking at transition state analogs.

The only way to know the transition state with sufficient accuracy is through computer simulations, such as those performed here.

In this work, we rationally identified mutations that increase the efficiency of the NCS, enabling its optimization for heterologous, or enzymatic, biosynthesis of the precursors of first-line analgesics. The proposed mutations are not disruptive from a structural perspective. The results point the way to the rational optimization of the crop-independent biosynthesis of BIAs.

## Data Availability

The data presented in this study can be found in the article or in the associated Appendix A.

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
