# Peer review of "Rational Engineering of (S)-Norcoclaurine Synthase for Efficient Benzylisoquinoline Alkaloids Biosynthesis"

_molecules, 2023, doi:10.3390/molecules28114265_

Round 1

Reviewer 1 Report

Nevertheless, the manuscript include only computational studies with known methodology, the Authors present very well how  application of QM/MM can be useful for analysis of biochemical mechanisms.

The manuscript is written at high scientific manner, probably checked in details by Authors, almost no mistakes found.

The conclusions are very clear, consistent with results and seem to have high scientific meaning.

I recommend this work to publish after minor corrections:

1) Author should check scheme 2 if the font size of drawn structure is the same, it looks like Lys121 (NH2 group is smaller?)

2) in the first sentence of Results and Discussion Authors write 'RMSd', shouldn't be 'RMSD'?   

Author Response

1) Author should check scheme 2 if the font size of drawn structure is the same, it looks like Lys121 (NH2 group is smaller?)

Answer: We thank the reviewer for their comment. In fact, all the letters on the left side of the scheme were in a smaller font. We have corrected this issue by increasing the letters font, in scheme 2.

2) in the first sentence of Results and Discussion Authors write 'RMSd', shouldn't be 'RMSD'?  

Answer: We thank the reviewer for their comment. This typo has been corrected.

Reviewer 2 Report

The manuscript "Rational Engineering of (S)-Norcoclaurine Synthase for Efficient Benzylisoquinoline Alkaloids Biosynthesis" by J. P. M. Sousa, N. C. S. A. Oliveira and P. A. Fernandes is devoted to the in silico analysis of point mutations which could be rate-increasing for  enzyme functionality.

The authors presented theoretical basis for rational enzyme improvement: they use complex model which include the molecular dynamics simulation. I guess the presented work is out of the current journal's scope due to the focus on molecular modelling. But it could be actual for another specialized journal like:

Journal of Molecular Graphics and Modelling;

International Journal of Quantum Chemistry;

Computational and Theoretical Chemistry;

Journal of Molecular Modeling etc.

Corrections needed:

1. Abstract: (S)-norcoclaurine... - (S)-Norcoclaurine (or higenamine)... ;

2. Please, carefully check the formatting. For example, the formatting inside the Abstract is not uniform, as well as References section has typed by another font (not like the main text). Formatting for Acknowledgement header looks like IT'S THE MAIN PART OF THE TEXT

3. Scheme 1: We could not detect the similarity between structures of codeine, morphine and thebaine. The cycles should have the same stereochemical designations in all structures due to its biosynthetic similarity. Also it's not clear why the other names are starting from capital letter but not the "norcoclaurine"?

4. page 7: Please, delete the text "(Error! Reference source not found.)." and make the right reference.

5. Please, correct the text for ref.2: "Organization, W. H., Who Model List of Essential Medicines–22nd List, 2021. Geneva, Who, 2021".

Author Response

The manuscript "Rational Engineering of (S)-Norcoclaurine Synthase for Efficient Benzylisoquinoline Alkaloids Biosynthesis" by J. P. M. Sousa, N. C. S. A. Oliveira and P. A. Fernandes is devoted to the in silico analysis of point mutations which could be rate-increasing for enzyme functionality.

The authors presented theoretical basis for rational enzyme improvement: they use complex model which include the molecular dynamics simulation. I guess the presented work is out of the current journal's scope due to the focus on molecular modelling. But it could be actual for another specialized journal like:

Journal of Molecular Graphics and Modelling;

International Journal of Quantum Chemistry;

Computational and Theoretical Chemistry;

Journal of Molecular Modeling etc.

Answer: We thank the reviewer for their comment and suggestions. In fact, molecular modeling methods are extensively used in this study, nonetheless, the focus of our work is to accurately describe the interaction of the quinonoid intermediate and transition state with the enzyme, thus providing valuable information for experimental campaigns targeting the improvement of (S)-norcoclaurine synthase.

Corrections needed:

  1. Abstract: (S)-norcoclaurine... - (S)-Norcoclaurine (or higenamine)... ;

Answer: We thank the reviewer for their comment. This typo has been corrected.

  1. Please, carefully check the formatting. For example, the formatting inside the Abstract is not uniform, as well as References section has typed by another font (not like the main text). Formatting for Acknowledgement header looks like IT'S THE MAIN PART OF THE TEXT

Answer: We thank the reviewer for their comment. The formatting has been made consistent throughout the manuscript.

  1. Scheme 1: We could not detect the similarity between structures of codeine, morphine and thebaine. The cycles should have the same stereochemical designations in all structures due to its biosynthetic similarity. Also it’s not clear why the other names are starting from capital letter but not the “norcoclaurine”?

Answer: We thank the reviewer for their comment. The structures of codeine, morphine, and thebaine have been redesigned and the stereocenter configuration was added to each stereogenic center to clearly evidence their structural and biosynthetic similarity. We have also enriched the information in the scheme by adding alternative pathways for the synthesis of NCS substrates. The typo regarding “norcoclaurine” has been corrected by starting its name with a capital letter.

  1. page 7: Please, delete the text "(Error! Reference source not found.)." and make the right reference.

Answer: The reference to Figure 3 has been added, thus fixing this issue.

  1. Please, correct the text for ref.2: "Organization, W. H., Who Model List of Essential Medicines–22nd List, 2021. Geneva, Who, 2021".

Answer: We thank the reviewer for its repair and we have corrected ref.2.

Reviewer 3 Report

The manuscript by Sousa et al. describes the results of computational prediction of (S)-norcoclaurine synthase (NSC) catalytic properties. Using the homology modeling, molecular docking, MD, and ONIOM methodologies, the authors identified the transition state of the key stage of the enzyme action and explored the possible mutations that would lead to better catalytic properties. The obtained results can be helpful for biotechnology production of benzylisoquinoline alkaloids and corresponding analgesics. The results are important and deserve publication.

Specific comments: 

Page 2, Line7 from top:  Please replace "phenilacetaldehide" with "phenylacetaldehyde". Check also throughout the manuscript.

Page 5, Paragraph 4 from top:  Although the authors give literature references concerning the "deleted residue" approach, an additional clarification of the methodology is necessary. Were the deleted residues replaced by some dummy fragments or were they just fully removed from the protein structure?

Page 7, the line before Figure 3:  The reference in the document is not defined.

Page 8, Line 4 from top:  Please replace "in the barrier" with "on the barrier".

Page 8, Line 16 from top:  Perhaps, the term "barrier stabilization" is not very suitable in this context.

Summarizing, I recommend acceptance of the manuscript for publication after minor revision.

Author Response

Page 2, Line7 from top:  Please replace "phenilacetaldehide" with "phenylacetaldehyde". Check also throughout the manuscript.

Answer: We thank the reviewer for their comment. We have corrected the typo.

Page 5, Paragraph 4 from top:  Although the authors give literature references concerning the "deleted residue" approach, additional clarification of the methodology is necessary. Were the deleted residues replaced by some dummy fragments or were they just fully removed from the protein structure?

Answer: We thank the reviewer for their comment. The residues were fully removed from the protein structure. To make this procedure clearer we added a new section with the details concerning the conceptual deletion experiment.

Page 7, the line before Figure 3:  The reference in the document is not defined.

Answer: We thank the reviewer for their comment. We have corrected this issue, as suggested by Reviewer 2.

Page 8, Line 4 from top:  Please replace "in the barrier" with "on the barrier".

Answer: We thank the reviewer for their comment. We have fixed the sentence as suggested by the reviewer.

Page 8, Line 16 from top:  Perhaps, the term "barrier stabilization" is not very suitable in this context.

Answer: We thank the reviewer for their comment. We agree with the reviewer and have replaced the “barrier stabilization” term with “barrier-lowering effect”, making it more suitable to classify the effect of Tyr60 in the barrier.

Round 2

Reviewer 2 Report

I'm not agree with the author's position that it's not a specific material. In my opinion the paper describing only molecular modelling is not suitable for multidisciplinary journal.

There's no any experimental support of the presented data. At least the amino acid sequences for analogous enzymes presented in the databases could be analyzed and compared with the proposed mutant enzymes.

And it's not clear for me - why the proposed mutations should be neutral for enzyme functionality. This point mentioned in the conclusion should be supported by data.

minor correction:

introduction, paragraph 2: formation of fusel products - why "fusel"? It's "side products".

I'm not agree with the author's position that it's not a specific material. In my opinion the paper describing only molecular modelling is not suitable for multidisciplinary journal.